# Saying “Yes” to NONO: A Therapeutic Target for Neuroblastoma and Beyond

**DOI:** 10.3390/cancers17193228

**Published:** 2025-10-03

**Authors:** Sofya S. Pogodaeva, Olga O. Miletina, Nadezhda V. Antipova, Alexander A. Shtil, Oleg A. Kuchur

**Affiliations:** 1Center for Molecular and Biological Technologies, ITMO University, 197101 Saint Petersburg, Russia; pogodaeva@itmo.ru (S.S.P.); miletina@scamt-itmo.ru (O.O.M.); 2Shemyakin-Ovchinnikov Institute of Bioorganic Chemistry, Russian Academy of Sciences, 117997 Moscow, Russia; 3Blokhin National Medical Research Center of Oncology, 115522 Moscow, Russia; shtil@scamt-itmo.ru; 4National Research Nuclear University MEPhI, 115409 Moscow, Russia; 5Higher School of Economics, 194100 Saint Petersburg, Russia

**Keywords:** gene transcription, core regulatory circuit, NONO, MYCN, long non-coding RNAs, therapeutic target, neuroblastoma, cancer

## Abstract

**Simple Summary:**

Neuroblastoma is a dangerous childhood cancer where many genes are in the “on” position, driving an uncontrolled tumor growth. A multifunctional non-POU domain-containing octamer-binding protein (NONO) acts as a master regulator of these genes. Therefore, NONO emerges as a key culprit of the disease’s aggressiveness and resistance to therapy. This review explores how targeting NONO could offer new therapeutic strategies. We discuss the challenges of drug development and highlight recent development of chemical tools to inhibit NONO function. Successful targeting of NONO is expected to provide powerful new treatments for children with high-risk neuroblastoma, potentially overcoming the limitations of current therapies.

**Abstract:**

Pediatric tumors such as neuroblastoma are characterized by a genome-wide ‘transcriptional burden’, surmising the involvement of multiple alterations of gene expression. Search for master regulators of transcription whose inactivation is lethal for tumor cells identified the non-POU domain-containing octamer-binding protein (NONO), a member of the Drosophila Behavior/Human Splicing family known for the ability to form complexes with macromolecules. NONO emerges as an essential mechanism in normal neurogenesis as well as in tumor biology. In particular, NONO interactions with RNAs, largely with long non-coding MYCN transcripts, have been attributed to the aggressiveness of neuroblastoma. Broadening its significance beyond MYCN regulation, NONO guards a subset of transcription factors that comprise a core regulatory circuit, a self-sustained loop that maintains transcription. As a component of protein–protein complexes, NONO has been implicated in the control of cell cycle progression, double-strand DNA repair, and, generally, in cell survival. Altogether, the pro-oncogenic roles of NONO justify the need for its inactivation as a therapeutic strategy. However, considering NONO as a therapeutic target, its druggability is a challenge. Recent advances in the inactivation of NONO and downstream signaling with small molecular weight compounds make promising the development of pharmacological antagonists of NONO pathway(s) for neuroblastoma treatment.

## 1. Introduction

In the growing list of molecular mechanisms critical in various aspects of cancer biology, gene transcription attracts increasing attention. The term ‘transcriptional burden’, coined by Comitani et al. as an alternative to the ‘mutational burden’ [1], highlights the significance of epigenetic regulation in specific malignancies, most importantly, in pediatric cancers. Along with a great amount of data on the pathogenetic role of transcriptional events in carcinogenesis, this mechanism is being considered a therapeutic target. Indeed, it is plausible to hypothesize that interference with this vital mechanism would be lethal for tumor cells.

Neuroblastoma, a malignancy that originates from embryonic neural crest cells, is one of the most common solid tumors in early childhood [2]. This aggressive cancer exhibits a broad spectrum of clinical manifestations, ranging from spontaneous regression to relentless progression despite multimodal therapies that include surgery, radiation, and high-dose chemotherapy [3,4]. Approximately 50% of cases are classified as high-risk with a five-year survival rate below 50% [5], underscoring the urgent need for novel therapeutic approaches.

Biological heterogeneity, a hallmark of neuroblastoma, is driven by epigenetic and, to a lesser extent, genetic alterations. Amplification of the *MYCN* oncogene is an established marker of neuroblastoma aggressiveness [6,7]. Among other genetic factors are the activating mutations in the ALK (anaplastic lymphoma kinase) gene that drive proliferation and the hemizygous deletion of the 11q chromosome associated with the loss of tumor suppressor genes [8,9,10].

More complex and pathogenetically significant are the genome-wide changes in the transcriptional landscape. Two major subtypes of neuroblastoma differ in the arrays of individual transcriptional mechanisms. The adrenergic (ADRN) subtype is characterized by *MYCN* amplification along with deregulation of a set of genes mechanistically linked to the MYCN network, including paired-like homeobox 2 (*PHOX2*), heart and neural crest derivatives-expressed protein 2 (*HAND2*), GATA binding protein 3 (*GATA3*), insulin enhancer binding protein 1 (*ISL1*), T-box transcription factor 2 (*TBX2*), and Achaete-scute family bHLH transcription factor 1 (*ASCL1*) genes [11,12,13]. The transcription factors encoded by these genes comprise a core regulatory circuit (CRC), a self-sustainable loop that maintains transcription of specific genes [10,13]. As a result of constitutively active transcription of CRC-dependent genes, the ADRN subtype is known for its aggressiveness. Unlike ADRN, the mesenchymal (MES) subtype contains CD133-positive cells that overexpress Yes-associated protein/transcriptional coactivator with PDZ binding motif (YAP/TAZ), paired related homeobox 1 (PRRX1), and Snail family transcriptional repressor 1 (SNAI1) proteins [9,14,15].

Amid the intricate transcriptional landscape, the non-POU domain-containing octamer-binding protein (NONO) emerges as a pivotal regulator. This multifunctional protein orchestrates gene expression by interacting with long non-coding RNAs (lncRNAs, transcripts > 200 bp) and transcription factors. Studies in normal cells and in tumor models, including neuroblastoma, position NONO as a bridge between genetic alterations and transcriptional reprogramming [16]. In neurogenesis, NONO overexpression during embryonic cortex development disrupts radial neuronal migration by impairing multipolar-to-bipolar transition—reducing uni/bipolar neurons and disorienting the leading processes relative to radial glia [17]. This leads to persistent postnatal disorders such as shortened processes, simplified dendritic arborization, and Golgi misorientation. These findings implicate dysregulated NONO in neurodevelopmental defects involving neuronal mispositioning. Since neuroblastoma is a transcription-dependent malignancy, one may hypothesize that NONO plays a substantial role in its pathogenesis. If so, can NONO be therapeutically druggable, alone and/or in combination with conventional agents?

## 2. NONO in Transcriptional Regulation

### 2.1. Physiological States

NONO belongs to the Drosophila behavior/human splicing (DBHS) family that also includes the paraspeckle component 1 (PSPC1) and the splicing factor, proline- and glutamine-rich (SFPQ) protein [18]. The NONO protein consists of four distinct regions: the RNA recognition motifs 1 and 2 (RRM1-2), the NONO/paraspeckle (NOPS), and the C-terminal coiled-coil domain. RRM1 and -2 are responsible for binding to RNAs, a function critically important for the role of NONO in RNA splicing and posttranscriptional regulation. These domains can dimerize upon protein–RNA complex formation [19]. The NOPS domain, being unique to the DBHS family, is necessary for the formation of paraspeckles, the membraneless dynamic organelles that control nuclear RNA retention and functions (reviewed in [20]). This domain allows for SFPQ-NONO heterodimerization [21]. The spiral domain at the C-terminus also participates in di- or oligomerization. Both N- and C-terminal regions are involved in phase separation to form condensates that concentrate the reaction components, thereby optimizing the conditions for their efficient interaction [22,23,24].

NONO, being part of macromolecular (i.e., protein–protein or protein–nucleic acid) complexes, orchestrates major cellular processes, namely, maintaining genome stability and gene expression landscape [18]. Structural data strongly suggest that NONO acts not as much as a single molecule. Two (perhaps not mutually exclusive) modes of transcriptional control by NONO imply (1) formation of complexes with RNA and (2) interactions with the transcription factors at the regulatory regions of target genes. First, NONO homodimers preferentially interact with RNAs in which adenosine is changed for inosine (A-to-I editing). This modification occurs in mRNAs, miRNAs, or non-coding transcripts. A-to-I editing can lead to interference with RNA splicing, resulting in nuclear accumulation of mRNAs [25,26]. The NONO/SFPQ/mat3 complex specifically binds to I-edited RNA, thereby entrapping it in the nuclei [27]. A pseudo-symmetric structure of NONO/PSPC1 allows for the heterodimer’s interaction with lncRNAs as well as with hyper-edited RNA sequences [19]. LncRNAs are especially important in neuroblastoma, being targets for MYCN oncoprotein (see section Relevance to neuroblastoma).

Second, NONO interacts directly with transcription factors to regulate developmental and cell cycle-related genes. In mouse embryonic stem cells, NONO binds Pax9, Tbx3, Cdx2, and Gata4 to control developmental loci. Besides transcription factors, NONO recruits and cooperates with chromatin modifiers. It physically associates with TET1—sharing up to 85% of ChIP peaks—to guide TET1 to neurogenic gene promoters. NONO knockdown displaced TET1, elevated 5-hydroxymethylcytosine at Bmp1, Fgf8, and Wnt3a genes, and down-regulated their expression [28]. Furthermore, NONO co-occupied chromatin with Erk2 and phosphoSer5 RNA polymerase II; Erk2 knockdown abolished NONO-chromatin association [29]. The role in chromatin remodeling is further supported by the interaction of NONO with the acetyltransferase CBP/p300 [30]. Moreover, NONO partners with TAZ to activate mechanoresponsive genes: following cell adhesion, NONO–TAZ complexes peak at 2–4 h, coinciding with TAZ dephosphorylation and driving CTGF/CYR61 expression. Over 80% of TAZ and ~50% of NONO ChIP-seq peaks are co-localized to active enhancers containing H3K4me1+, H3K4me3− or H3K27Ac+ epigenetic marks. Loss of NONO significantly reduced TAZ distribution across the enhancer regions [31].

Studies in primary mouse fibroblasts showed that, while circadian rhythms remained intact in NONO-deficient cells (via gene trap technology—NONO gt), the circadian control of G1 exit was lost: cells proliferated faster, p16Ink4A expression decreased, senescence markers were reduced, and the circadian gating of S-phase entry was suppressed. At the molecular level, NONO interacts with circadian PERIOD (Per1/Per2) proteins, co-activating the p16Ink4A gene and enforcing temporal segregation of the cell cycle. Deletion of either NONO or PERIOD disrupted the rhythmic expression of p16Ink4A and abolished circadian gating of cell division. The absence of NONO resulted in wound healing defects similar to those observed in mice with disrupted circadian mechanisms [32].

Intriguingly, physical interaction of NONO with protein partners does not necessarily presume transcriptional activation. Dong et al. [33]. demonstrated that, in rat myometrium cells, NONO forms complexes with progesterone receptors via direct binding independently of the hormone. These complexes are recruited to the *connexin 43* (*Gja1*) promoter to attenuate this gene. Accordingly, siRNA-mediated NONO knockdown elevated *Gja1* transcripts 2.5-fold. Transcriptomic analysis further substantiated the significance of NONO as a repressor of 150 genes associated with myometrial contractility. This study places NONO into the regulatory network that controls the timing of labor.

Together, the ability to interact with nucleic acids and proteins makes NONO a versatile regulator of gene expression in a variety of physiological situations. It is plausible to expect that this role is translated to (or reflected in) tumor biology.

### 2.2. Roles in Cancer

As in non-malignant cells, functions of NONO in cancer are implemented via RNA binding as well as protein–protein complex formation. NONO has been implicated in the progression of colorectal cancer due to its interaction with GAPLINC (gastric adenocarcinoma-associated, positive CD44 regulator, long intergenic non-coding RNA). Knockdown of NONO attenuated GAPLINC-activated invasion of HCT116 colon carcinoma cells in Transwell assays. Global analysis of gene expression profiles showed that NONO knockdown decreased the expression of 1776 genes that overlapped with GAPLINC-induced *SNAI2*, *LAMB3,* and *IGFBP7* genes whose functions are linked to metastasis [34]. Similarly, Cheng et al. [35] showed that increased NONO levels correlated with the depth of invasion in esophageal squamous cell carcinoma (ESCC); NONO played a critical role in suppressing apoptosis through activation of Akt and Erk1/2 signaling pathways. NONO silencing by siRNA inhibited ESCC cell proliferation and invasion while inducing apoptosis, indicating its key contribution to ESCC tumorigenesis [35].

In hepatocellular carcinoma, NONO interacts with lncRNA DIO3OS, an anti-oncogenic transcript down-regulated in cancers. DIO3OS implements its suppressive role through down-regulation of zinc finger E-box binding homeobox 1 protein (ZEB1), a factor of tumor cell stemness. Binding of DIO3OS to NONO restricts nuclear export of ZEB1 mRNA. Therefore, the DIO3OS-NONO-ZEB1 axis emerges as a mechanism of control of malignant potential in hepatocellular carcinoma [36].

Functions of NONO as an RNA binder and a component of protein–protein complexes are not mutually exclusive but rather act in concert. In U2OS sarcoma cells, DNA damage triggered NONO relocation from gene promoters to the nucleoli, an effect relevant to the attenuation of de novo pre-mRNA synthesis for rescuing the transcripts from degradation. In untreated cells, NONO formed stable complexes with RNA polymerase II. These complexes were largely disrupted in response to topoisomerase II poison etoposide. NONO inhibition led to an increase in γH2AX foci and acetylated histone marks (H2BK120ac) on the transcription start sites of actively expressing genes. NONO recruited X-ray repair cross-complementing protein 4 (XRCC4) for double-strand DNA repair via a non-homologous end joining; NONO deficiency prevented the repair [37].

The response of NONO to genotoxic stress was further dissected by Li et al. [38]. The authors subjected mouse males to whole-body ionizing radiation and found that NONO levels in testicles were elevated severalfold by 24–48 h post exposure to 4 Gy. NONO deficiency enhanced radiation-induced apoptosis in Sertoli cells, supporting the positive role of NONO in double-strand DNA repair and rescue of testicular cells. Interestingly, SFPQ tended to compensate for NONO inactivation because irradiation of gt/0 males, but not wild-type counterparts, led to an increase in SFPQ [38]. Furthermore, NONO silencing altered intra-S-phase checkpoint activation in UV-irradiated HeLa and melanoma cells, evidenced by continued DNA synthesis, failure to block origin firing, and impaired CHK1S^345^ phosphorylation. NONO re-expression rescued checkpoint function [39]. These findings are in line with other studies of NONO participation in DNA damage repair [40,41].

In breast cancer cells, NONO acts as a component of chromatin-remodeling machinery in complexes with EGFR and STAT3. Accordingly, the pro-proliferative and pro-survival genes are activated: *Aurora A*, *B-myb*, *c-Myc* (EGFR targets), and *CCNB1* and *CCND1* (STAT3 targets) [30,42]. Interaction of NONO with sterol regulatory element-binding proteins SREBP-1a and SREBP-2 increased the intracellular amounts of triglycerides > 3-fold. Knockdown of NONO negatively regulated the abundance and accumulation of cholesterol [43].

Among post-translational mechanisms of NONO regulation, evidence is growing in support of proteolysis-associated events [44]. One may expect that NONO, as a partner in protein–protein complexes, should be protected from proteolytic degradation. Feng et al. [45] demonstrated that, in melanoma cells, the ubiquitin-specific protease 11 (USP11) interacted with nuclear NONO and prevented its subsequent proteolysis by hydrolyzing the polyubiquitinated chains. USP11 deficiency slowed down melanocyte proliferation; cell growth rate was restored with the exogenous NONO, suggesting that NONO is a pro-proliferation factor. A positive correlation between the amounts of USP11 and NONO was confirmed in clinical melanoma samples [45]. Figure 1 shows the main functions of NONO.

### 2.3. Relevance to Neuroblastoma

Thus, NONO emerges as a genome-wide guardian of transcription. This pivotal role is implemented via the formation of complexes with other proteins and lncRNAs to ensure the onset of transcription and message stabilization. High NONO expression in neuroblastoma correlates with elevated MYCN levels and poor patient survival. In 88 out of 341 patient samples, *MYCN* amplification was associated with lncUSMYCN overexpression [46,47]. Interaction of lncUSMYCN RNA with NONO activates NCYM, an antisense transcript that prevents MYCN ubiquitination by inhibiting its GSK3β-mediated degradation [46]. Thus, NONO promotes MYCN-driven oncogenesis. Antisense lncUSMYCN oligonucleotides may therefore be considered as tools to abrogate NONO binding for limiting neuroblastoma progression in mouse models.

Along with lncMYCN, the MYCN protein binds other neuroblastoma-specific lncRNAs such as CASC15, LINC00511, ZRANB2-AS2, and PPP1R26-AS1; the former two transcripts have been characterized as risk factors [48]. The neuroblastoma-associated transcript 1 (NBAT-1) lncRNA functions as a tumor suppressor by down-regulating the genes involved in proliferation and invasion, e.g., *SOX9*, whose overexpression impedes retinoic acid-induced neuronal differentiation. NBAT-1 lncRNA overexpression contributes to inhibition of neuroblastoma growth and metastasis; therefore, efforts have been put on activation of this transcript [49,50,51]. Accordingly, NBAT-1 lncRNA levels inversely correlated with MYCN, suggesting that NBAT-1 lncRNA can be down-regulated in the aggressive disease [52]. NONO binds to lncUSMYCN RNA, leading to *MYCN* up-regulation and proliferation of neuroblastoma cells [47,53].

**Figure 1 cancers-17-03228-f001:**
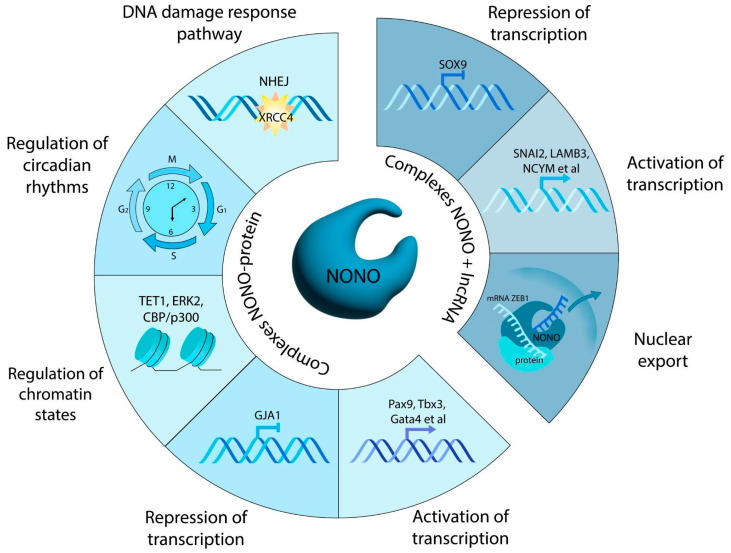
Effects of the multifunctional protein NONO. Summarized are NONO functions in the context of protein–protein and protein–RNA complexes, as well as molecular mechanisms underlying each function.

Two lncRNA isoforms, NEAT1_1 and NEAT1_2, are also being investigated as NONO partners in neuroblastoma. Elevated NEAT1_2 promotes the formation of NONO-containing paraspeckles. The increase in NEAT1_2 with anti-NEAT1_1 oligonucleotides led to sequestration of NONO in paraspeckles. This effect can limit the accessibility of NONO to pro-oncogenic lncRNAs, e.g., lncUSMYCN, thereby decreasing tumor cell viability and/or activating differentiation. Consequently, the anti-oncogenic NONO–lncNEAT1 interaction may attenuate neuroblastoma aggressiveness [20,54]. The study by Simko et al. [55] revealed that G-quadruplex (G4) structures in lncRNA NEAT1 serve as a conserved structural motif for recruiting NONO, a critical step in paraspeckle assembly. The authors demonstrated that NONO exhibits structural specificity for G4 conformations—initially identified in C9orf72 repeat RNA—and that NEAT1 lncRNA harbors abundant, evolutionarily conserved G4 motifs. The persisted G4 enrichment positions the RNA quadruplexes as primary structural elements mediating NONO recruitment and paraspeckle biogenesis.

Yet another level of complexity is provided by recent insights into NONO-dependent gene regulation in neuroblastoma via the formation of nucleic acid-containing condensates across the nuclei. Phase separation allows for NONO involvement in super-enhancer-regulated *HAND2* and *GATA2* genes whose products modulate neural cell differentiation and migration [56,57,58].

Furthermore, in neuroblastoma cells, NONO acts as a partner with transcription factors. NONO and SOX2 proteins form a bidirectional regulatory network that influences neuroblastoma heterogeneity, stem cell-like properties, plasticity, and resistance to therapy [59]. NONO acts as a transcriptional silencer of the *SOX2* gene in neuroblastoma cells. NONO binds to the *SOX2* promoter, repressing its transcription and contributing to the maintenance of stemness [60]. In turn, SOX2 protein represses the *NONO* gene through a feedback loop, potentially limiting NONO-mediated oncogenic transcription [61].

Along with SOX2, the SOX11 transcription factor emerges as a pivotal regulator in ADRN neuroblastoma, where its amplification stabilizes a lineage-specific CRC essential for tumorigenesis. This effect is mediated by physical interactions with DBHS proteins, enabling SOX11 to modulate alternative splicing while co-regulating the transcription of splicing factors SRSF2, RBMX, and RBM4. Partnership of NONO with SOX11 enhances mRNA processing of super-enhancer-driven oncogenes, including CRC members *GATA2* and *HAND2*. The SOX11-NONO axis links chromatin remodeling and RNA splicing, creating a coherent epigenetic RNA operon that confers ADRN identity and represents a therapeutic vulnerability in *MYCN*-amplified tumors [62]. This reciprocal regulation suggests that the NONO-SOX2/SOX11 axes are critical for neuroblastoma progression via the balance between stemness and differentiation.

In complexes with YY1 transcription factor, NONO regulates the expression of the *PARP1* gene, whose product covalently adds poly(ADPriboso) moieties to a variety of targets, including NONO. PARylation stabilizes NONO by protecting it from ubiquitination. The YY1-PARP1-NONO axis maintains the maturation of *ADAM8* and *TEX14* transcripts associated with aggressive disease and poor survival [63]. This mechanism, together with YY1-mediated activation of glycolysis genes [64], further supports the negative role of this transcription factor and implicates NONO in YY1-dependent phenomena in neuroblastoma.

Overall, NONO maintains the pro-oncogenic phenotypes in the ‘transcriptional’ tumor, such as neuroblastoma via the formation of complexes with nucleic acids and transcription-related proteins (Figure 2). These remarkable abilities justify NONO as a therapeutic target.

## 3. Therapeutic Potential of NONO Targeting: Not Only Neuroblastoma

Roles of NONO in anticancer drug response, as well as the opportunities to pharmacologically inactivate this mechanism, deserve a thorough investigation. At present the reports on these potentially exciting problems are scarce. Tsofack et al. [65] have found that NONO interacts with the YB-1 transcription factor in SW480 and HT29 colon carcinoma cell lines. Knockdown of NONO decreased the levels of YB-1 mRNA and protein twofold, in parallel with cell sensitization to oxaliplatin. In turn, YB-1 knockdown attenuated NONO abundance. Overexpression of YB-1 decreased the response to oxaliplatin 2.5–4.4-fold, whereas NONO deficiency sensitized YB-1-overexpressing cells to this drug up to 4.4-fold after 16 h exposure. These results provide strong evidence in favor of the NONO-YB-1 partnership in transcriptional control of survival of colon cancer cells treated with the platinum-containing drug [65].

Cytosine arabinoside is an essential component of therapeutic regimens in acute myelogenous leukemia (AML). NONO stabilized the deoxynucleoside triphosphate triphosphohydrolase SAMHD1 by preventing its ubiquitination. Elevated levels of SAMHD1 were mechanistically attributed to the resistance of AML cells to cytosine arabinoside; down-regulation of NONO sensitized cells to this drug via decreasing SAMHD1. Interestingly, DNA-damaging drugs DDP and adriamycin were able to decrease NONO and SAMHD1 proteins, thereby increasing the efficacy of cytosine arabinoside against otherwise resistant AML cells [66,67].

In triple-negative breast cancer cell lines (Hs 578T, MDA-MB-231, MDA-MB-468), NONO knockdown (shRNA) suppressed proliferation and colony formation. In MDA-MB-231-derived xenotransplants, siRNA NONO attenuated Ki-67 expression and tumor growth. Li et al. [68] reported that NONO, together with PSPC1 and SFPQ, participates in double-strand DNA repair. Knockout of NONO was tolerated by mouse embryo fibroblasts irradiated with γ-photons. However, radiosensitivity was potentiated by knocking out the *PSPC1* gene coupled with the *NONO* knockout. These results substantiated the search for pharmacological blockers of NONO protein.

Screening of FDA-approved drugs identified auranofin as a NONO antagonist. This agent caused a decrease in NONO mRNA and protein in MDA-MB-231 cells [42]. Floros et al. [69] demonstrated that auranofin (0.33–3.3 μM) was cytotoxic for *MYCN*-amplified cell lines (IMR-5, SK-N-DZ, Kelly) but not for SK-N-SH and CHLA20 cells that carry a single *MYCN* gene copy. Ferrostatin-1 attenuated the cytotoxicity of auranofin in Kelly and SK-N-DZ cell lines, supporting a role for ferroptosis in auranofin-induced cell death. Administration of auranofin (10 mg/kg i.p. daily) abrogated the growth of IMR-5 transplants as well as patient-derived xenografts in NCG mice. This compound elevated ferroptosis markers in IMR-5 tumors [69]. However, auranofin-induced NONO decrease might be indirect. In our experiments, the combination of auranofin with siRNA-mediated *NONO* knockdown potentiated the reactive oxygen species (ROS) burst and death of the MYCN-amplified Kelly cell line [70]. If NONO is indeed a redox sensor, its inhibition justifies the strategy of enhancing the efficacy of ROS-generating drug regimens. This mechanism might be particularly important for the eradication of cells that acquired pleiotropic resistance in the course of consecutive rounds of chemotherapy [71].

In *MYCN*-amplified tumors, dual inhibition of PARP and CHK1 intensified replication stress and blocked S-phase checkpoint control, causing unresolved DNA damage and mitotic catastrophe. PARP blockade stalls replication forks, while CHK1 inhibition prevents their stabilization and homologous recombination repair. This synthetic lethality is selectively potent for MYCN-driven neuroblastoma cell lines (IMR-32, LAN-5, SH-EP). In xenograft models, a low-dose PARP inhibitor, olaparib, plus a CHK1 blocker, MK-8776, markedly suppressed tumor growth and prolonged survival without significant systemic toxicity [72]. Given that NONO is a part of DNA repair machinery, it is worth testing whether NONO depletion would further decrease the viability of aggressive neuroblastoma cells in combination with PARP1/CHK1 inhibition.

The above considerations make NONO inhibition therapeutically meaningful. Indeed, its down-regulation would be beneficial for simultaneous attenuation of several mechanisms. However, the structure of NONO provides a serious challenge for its druggability. A significant step forward has been made by Kathman et al. [73]. The authors screened a broad library of small molecular weight compounds in search of an agent capable of countering the function of the androgen receptor (both wild type and splice variants) in prostate carcinoma cells. The compound termed (R)-SKBG-1 covalently interacted with the cysteine residue at position 145 of NONO. This site appeared to be critical since NONO disruption or cysteine-to-serine substitution at this position abrogated cell sensitivity to (R)-SKBG-1. Furthermore, NONO inactivation was stereoselective because the (S)-SKBG-1 isomer was inefficient. Importantly, (R)-SKBG-1 did not cause NONO depletion but instead trapped the protein to RNA, thereby stabilizing the complex. As a net effect of (R)-SKBG-1, the abundance of androgen receptor mRNA substantially decreased. This elegant study convincingly demonstrated that (1) NONO is an upstream regulator of the gene pathogenetically involved in carcinogenesis and (2) NONO is a druggable target.

It is plausible to hypothesize that pharmacological inactivation of NONO can be therapeutically relevant in neuroblastoma. With the pioneer chemical instrument (R)-SKBG-1 and its derivatives being designed on the basis of the crystal structure of ligand–NONO complexes [74,75], it becomes possible to address the question of whether NONO inhibition would down-regulate a single or a couple of NONO-dependent CRC genes. In turn, the combinations of conventional chemotherapeutics with (R)-SKBG-1 are expected to reveal the potential of NONO inactivation as a component of treatment regimens if inactivation of NONO alone is insufficient for killing tumor cells.

An imidazolium-based compound, sepantronium bromide (YM155), has been recently reported to prevent the formation of complexes between the interleukin enhancer binding factor 3 (ILF3) and NONO (p54nrb), a mechanism involved in the *survivin* gene regulation. In HEK293 and PC-3 cell lines, YM155 down-regulated this gene via destabilization of ILF3/p54nrb complexes. In co-immunoprecipitation experiments, this compound decreased IFL3–p54nrb interaction by ~80% without affecting the abundance of each protein. ChIP analysis revealed that p54nrb and ILF3 were enriched on the *survivin* gene promoter, whereas YM155 attenuated these interactions. The siRNA-assisted p54nrb knockdown led to down-regulation of the *survivin* gene [76]. YM155 entered clinical trials in several intractable tumors, including castration-resistant prostate cancer. Thus, this agent could be useful for the design of therapeutic regimens in MYCN/NONO-overexpressing neuroblastoma. One may expect that YM155 will be efficient in combinations with another chemotherapeutic(s). However, the chemical structures of R(SKBG-1) and YM155 are strikingly different, surmising that each compound acts via an individual mechanism. On the one hand, this diversity expands pharmacological opportunities for NONO inactivation. On the other hand, the precise mechanisms remain elusive due to manifold modes of NONO regulation.

Finally, regulation of NONO ubiquitination should be considered therapeutically relevant. As mentioned in Section 2.2, targeting USP can be an option for the control of NONO stability to proteasomal degradation. Inhibition of USP7 and −28 destabilized MYCN in SK-N-BE(2) and IMR32 cell lines [77,78]. USP11 in complexes with the transcription elongation factor A-like 1 has been found to be essential for the expression program of mesenchymal genes [79]. Lone et al. reported an indirect way of NONO destabilization by inhibiting its partner peptidyl-prolyl *cis*-*trans*-isomerase NIMA-interacting 1 (Pin1) protein with 5-hydroxy-1,4-naphthalenedione (Juglone) [44]. This effect was reversible by the proteasomal blocker MG-132. Since NONO regulation by small molecular weight compounds is a developing area, the selectivity and the mechanisms warrant a cautious interpretation. Table 1 summarizes information on NONO antagonists applicable in neuroblastoma and other tumors.

## 4. Conclusions

It is the genome-wide disordering of transcription that underlies a remarkable biological heterogeneity of neuroblastoma. The current classification determines transcriptional mechanisms unified by common regulation. According to this analysis, the ADRN and MES subtypes are characterized by the sets of 18 and 20 transcription factors, respectively [15]. Some of these factors comprise a specific autoregulatory loop termed CRC, a hallmark of tumors with ‘transcriptional’ (rather than ‘mutational’) burden [1]. This machinery forms a network of intergenic interactions that are ultimately translated into particular clinical manifestations [80].

It is logical to suggest that this complex machinery is governed by the hierarchy of mechanisms. Among other candidates, the NONO protein emerged as a major master regulator of specific genes, including CRC. Not surprisingly, NONO has been implicated in a plethora of functions in normal development as well as in tumor biology, particularly in the neural tissue. Now the question arises as to whether such a multifunctional mechanism should—or may—be considered a therapeutic target?

Several challenges remain before we can definitively target NONO therapeutically. First, as a protein that lacks the enzymatic function, NONO is expected to be hardly druggable since there is no real opportunity to employ structural features amenable for chemical targeting. Moreover, if NONO acts not as much as a homodimer but in partnership with other DBHS family proteins and specific RNAs, its druggability becomes even more problematic. The discovery of the small molecular weight covalent NONO blocker (R)-SKBG-1 set the stage for systematic investigation of mechanisms of NONO inactivation. Intriguingly, in (R)-SKBG-1-treated cells, the abundance of NONO remained unaltered, whereas NONO-dependent expression of the androgen receptor gene was abolished [73]. The mechanism postulates that (R)-SKBG-1 interferes with NONO–RNA interaction. Detailed structural investigation of ligand–NONO complexes requires molecular modeling and X-ray studies; also, dissection of structure–activity relationship by analyzing individual chemical moieties in (R)-SKBG-1, e.g., the reactive chloroacetyl group, is worthy. On the other hand, can NONO and its partners be inactivated with molecular glues, a developing class of compounds aimed at sticking to and fixing the non-enzymatic proteins (see [81] for a recent review)? Is there a way for PROTAC technology to proteolytically eliminate NONO [82,83]? Intriguingly, the net effects of NONO inactivation by each of the three approaches (i.e., covalent binding of the small molecular weight compound vs. the molecular glue vs. PROTAC) may differ. In other words, the presence of inactive NONO in the cell may or may not replicate the outcome of its absence.

Next, pharmacological inactivation of NONO alone, even if the target genes are attenuated, may or may not be sufficient for cell death. The effect of NONO inhibition on cell survival would depend on the significance of individual NONO-dependent genes for a particular cell type in a specific context. This notion holds true for relapsed neuroblastoma after the repeated rounds of chemo- and radiotherapy. What happens to NONO in the course of prolonged treatment, and what is an additional mechanism(s) suitable for drug combinations? The former question needs special investigation; for the latter opportunity, the literature data points at DNA repair proteins as tentative candidates. The availability, tolerability, and efficacy of inhibitors of the DNA damage response pathway make their combinations with the NONO blockers straightforward.

Of the utmost importance are the translational aspects of NONO deregulation. These problems are expected to be addressed with small molecular weight tools, in particular, (R)-SKBG-1 as a prototype. Specific prioritization of NONO-governed signaling axes in ADRN and MES phenotypes, the significance of NONO (and perhaps other DBHS proteins) as a biomarker for the therapeutic regimen, as well as the emergence of drug resistance and its circumvention, all may become investigational subjects for practical purposes not before the pharmacological agents for targeting ‘hardly druggable’ mechanisms are validated.

In summary, the strategies of targeting the master regulators in ‘transcriptional’ tumors, including neuroblastoma, are founded on the growing knowledge of molecular pathogenesis of gene expression alterations. One may expect that inactivation of one druggable mechanism would cause a ramification effect, that is, inhibit a number of downstream pathways. Trials of the current NONO antagonists in monotherapy and in combinations with conventional and targeted drugs will demonstrate whether the antitumor efficacy is achievable within a reasonable therapeutic ‘window’.


** **



**Note added in proof**


When this study was under review, a paper substantiating the role for NONO in metabolism was published [84]. The authors demonstrated that NONO inactivation by siRNA or (R)-SKBG-1 decreased the level of cholesterol in patient-derived cell lines, including Kelly. Together with [43], this important finding establishes the significance of NONO in transcriptional regulation of metabolism in neuroblastoma.

## Figures and Tables

**Figure 2 cancers-17-03228-f002:**
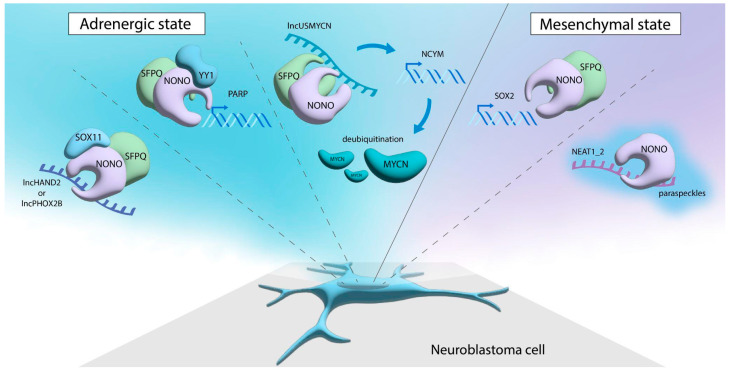
NONO in neuroblastoma states. Shown are interactions of NONO with DBHS family proteins or RNA in ADRN and MES subtypes. Note that these interactions are not strictly subtype specific. See text for details.

**Table 1 cancers-17-03228-t001:** Small molecular weight compounds as NONO antagonists.

Compound	Mechanism of Action/Effects	Trial	Refs.
**R-SKBG-1** 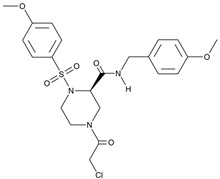	Covalent inhibitor Binds Cys145 to lock NONO in an RNA-bound state for inactivation of androgen receptor gene regulation	Preclinical	[73,74,75] *
**Auranofin** 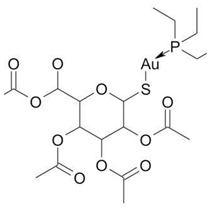	Pro-oxidant Suppresses NONO (possibly through thioredoxin reductase inhibition) to enhance ROS-driven ferroptosis in *MYCN*-amplified cells	Phase II: chronic lymphocytic leukemia (NCT01419691) Phase I: non-small cell lung carcinoma or small cell lung carcinoma (NCT01737502), ovarian, primary peritoneal, or fallopian tube cancer (NCT01747798)	[42,71,72] *
**Sepantronium bromide (YM155)** 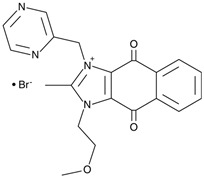	Gene expression Disrupts NONO–ILF3 complex on the *survivin* promoter, thereby down-regulating this gene and inducing apoptosis	Phase I/II: solid tumors, NSCLC (NCT01100931); Phase II: melanoma (NCT01009775)	[76]
**Olaparib + MK-8776** 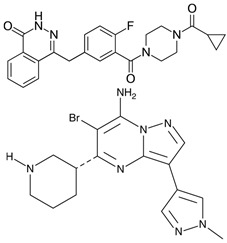	DNA repair Olaparib inhibits PARP1-mediated DNA repair, whereas CHK1 antagonist MK-8776 blocks DNA damage response kinase. This combination is supposed to promote replicative stress in cells with inactive NONO	Olaparib—Phase II: BRCA-mutated tumors (NCT04041128, NCT02677038) MK-8776—Phase I: solid tumors, lymphoma (NCT00779584)	[72]
**Juglone** 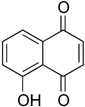	Protein lifespan Inhibits Pin-1-dependent NONO stabilization	Preclinical	[44]

* Note that [70,74,75] refer to the sources that have not yet been peer-reviewed.

## Data Availability

Data sharing is not applicable.

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
