# Peer review of "Saying “Yes” to NONO: A Therapeutic Target for Neuroblastoma and Beyond"

_cancers, 2025, doi:10.3390/cancers17193228_

Round 1

Reviewer 1 Report

Comments and Suggestions for Authors

"Saying 'Yes' to NONO: A Target for Neuroblastoma Therapy?"

Recommendation: Accept with Minor Revisions

Pogodaeva et al. present NONO protein as a promising neuroblastoma therapeutic target. The manuscript details NONO's fundamental roles in physiological and and cancer contexts, emphasizing its critical functions in neuroblastoma. The authors synthesize literature, arguing that NONO, a master transcription regulator in a "transcriptionally-addicted" cancer, is a high-value target. The discussion of NONO's involvement in the core regulatory circuit and its interactions with MYCN-associated long non-coding RNAs is particularly strong. The manuscript highlights therapeutic potential while acknowledging challenges in targeting a non-enzymatic protein.

The conclusion accurately states NONO is "hardly druggable" due to its interaction with macromolecules. Briefly elaborate on the difficulties of targeting large, flat, and dynamic protein-protein or protein-RNA interfaces to highlight the significance of identifying a covalent inhibitor like (R)-SKBG-1.

The review lacks critical discussion of methodological strengths and limitations in cited experimental studies (e.g., cell line models, in vivo relevance, reproducibility). A more explicit appraisal of models, such as the sufficiency of xenograft studies or accuracy of cell line models in mimicking MYCN biology, would improve depth.

Abstract: Remove unneeded hyphenation “pro-tein-protein”, “con-sidering“

Introduction: This sentence feels misplaced. Move earlier in this paragraph. “Resistance to chemotherapy is a characteristic trait of this immature (stem-like) 78 phenotype.” 

Move 17 ref citation earlier right after the first sentence discussing the role of NONO in cortex development.

What symbol is intended between EGFR and STAT3? “In breast cancer cells, NONO acts as a component of chromatin-remodeling machin-ery in complexes with EGFR и STAT3. “ This appears to be a russian character. Please replace character.

Consider citing a reference paper or review highlighting the role of antisense strategies for this sentence “Antisense lncUSMYCN 229 oligonucleotides may therefore be considered as tools to abrogate NONO binding for lim-iting neuroblastoma progression in mouse models.”

While the paragraph on pg8 line 340-349 is interesting, it's largely speculative as there are no data connecting this dual targeting strategy to potentiate NON depletion leading to replication stress. Rephrase this paragraph as a potential avenue for future investigation.

No discussion of synergy between pharmacological NONO inhibitors are discussed (e.g., auranofin, (R)-SKBG-1) and PARP/CHK1 inhibitors. Consider discussing the implications of their use and viable non-toxic combinations in MYCN-amplified neuroblastoma based on existing efforts in other tumor systems and tolerability in Phase 1 studies.

Do the authors expect chronic exposure of NONO-inhibited cells to chronic low-dose chemotherapeutic stress to identify resistance mechanisms (e.g., upregulation of SFPQ, PSPC1)?

Are there data evaluating the use of patient-derived xenografts (PDX) with varying NONO expression to test efficacy of NONO-targeting drugs and correlate with transcriptional signatures?

While pharmacological avenues appear to be promising means of targeting NONO, some discussion about how future investigations should strengthen the authors arguments regarding the role of NONO would be beneficial rather than jumping straight to pharmacological strategies. The authors allude to this issue in lines 435-444, by indicating that this may not be sufficient to yield NB cell death. As a result, some suggested experiments which explore the role of NONO would be beneficial. I.e. could the use of CRISPR-Cas9 be beneficial to selectively deplete NONO function in neuroblastoma cell lines (both MYCN-amplified and non-amplified) would allow for researchers to assess viability, transcriptomic shifts, and MYCN target gene expression?

Replace or expand figure1 with a schematic that summarizes all known NONO interactions in neuroblastoma, including lncRNA binding, protein-protein interactions, its role in DNA repair, and potential therapeutic intervention points. Another option would be a supporting table which summarizes some of the key functionality and supporting references that are used elsewhere in the review.

Improve Table 1 by adding further mechanistic information (e.g., covalent inhibitor, redox modulator, PARP pathway interference).

Misspelling: first sentence of section 3, the word "pharmacologically" is misspelled as "pharmacalogically." 

Author Response

We are grateful to the reviewer for interest and thoughtful comments.

Comment 1: Pogodaeva et al. present NONO protein as a promising neuroblastoma therapeutic target. The manuscript details NONO's fundamental roles in physiological and cancer contexts, emphasizing its critical functions in neuroblastoma. The authors synthesize literature, arguing that NONO, a master transcription regulator in a "transcriptionally-addicted" cancer, is a high-value target. The discussion of NONO's involvement in the core regulatory circuit and its interactions with MYCN-associated long non-coding RNAs is particularly strong. The manuscript highlights therapeutic potential while acknowledging challenges in targeting a non-enzymatic protein.

The conclusion accurately states NONO is "hardly druggable" due to its interaction with macromolecules.

Briefly elaborate on the difficulties of targeting large, flat, and dynamic protein-protein or protein-RNA interfaces to highlight the significance of identifying a covalent inhibitor like (R)-SKBG-1.

These considerations are briefly analyzed on page 13, Conclusions. Indeed, difficulties arise from the fact that NONO is a component of protein-protein or RNA-protein complexes, so it is problematic to design a chemical compound that disrupts the macromolecular interfaces. Furthermore, NONO lacks enzymatic function, therefore there is no chance to utilize specific structural pockets, a strategy widely used in targeting protein kinases or other bioenzymes. The covalent ligand (R)-SKBG-1 is a novel significant step forward in medicinal chemistry of NONO inhibitors.

Comment 2: The review lacks critical discussion of methodological strengths and limitations in cited experimental studies (e.g., cell line models, in vivo relevance, reproducibility). A more explicit appraisal of models, such as the sufficiency of xenograft studies or accuracy of cell line models in mimicking MYCN biology, would improve depth.

We agree that each of the above mentioned approaches has intrinsic advantages and limitations. This holds true not solely for neuroblastoma research but for experimental biology in general. In our analysis we tried to critically assess every source, keeping in mind data soundness, reproducibility, statistics, etc. Therefore, the review is based on a rather big number of citations to present the current state-of-the-art in the field. No doubt, the results with cell lines or xenografts or cell-free systems, however convincing they are within the experiment, must not be interpreted as direct indication for clinicians. At present the problem is to uncover molecular pathogenesis of neuroblastoma. In this sense we critically assessed the mechanistic role of auranofin in NONO inactivation. Overall, we keep practical recommendations as cautious as possible.

Comment 3: Abstract: Remove unneeded hyphenation “pro-tein-protein”, “con-sidering“

I see no hyphenation in these words. Probably they appeared during Word-to-pdf conversion.

Comment 4: Introduction: This sentence feels misplaced. Move earlier in this paragraph. “Resistance to chemotherapy is a characteristic trait of this immature (stem-like) 78 phenotype.” 

We discarded this sentence. The problem of therapeutic resistance is too complex to be mentioned briefly. We think such a superficial statement can be misleading.

Comment 5: Move 17 ref citation earlier right after the first sentence discussing the role of NONO in cortex development.

We moved [17] after the word ‘glia’. 

Comment 6: What symbol is intended between EGFR and STAT3? “In breast cancer cells, NONO acts as a component of chromatin-remodeling machin-ery in complexes with EGFR и STAT3. “ This appears to be a russian character. Please replace character.

Done

Comment 7: Consider citing a reference paper or review highlighting the role of antisense strategies for this sentence “Antisense lncUSMYCN oligonucleotides may therefore be considered as tools to abrogate NONO binding for limiting neuroblastoma progression in mouse models.”

This sentence concludes the two previous ones [47, 48]. A new ref. is unnecessary.

Comment 8: While the paragraph on pg8 line 340-349 is interesting, it's largely speculative as there are no data connecting this dual targeting strategy to potentiate NONO depletion leading to replication stress. Rephrase this paragraph as a potential avenue for future investigation.

This paragraph indeed points at the potential of NONO inhibition strategies. At the moment it is premature to state that combinations of DNA repair blockers and NONO antagonists are efficient since the pharmacological tools for the latter target have been tested only in cell culture. The final sentence in this paragraph is our hypothesis.

Comment 9: No discussion of synergy between pharmacological NONO inhibitors are discussed (e.g., auranofin, (R)-SKBG-1) and PARP/CHK1 inhibitors. Consider discussing the implications of their use and viable non-toxic combinations in MYCN-amplified neuroblastoma based on existing efforts in other tumor systems and tolerability in Phase 1 studies.

Combinations of auranofin and (R)-SKBR-1 are exactly the subject of our current experiments. We see a nice synergy in MYCN-amplified neuroblastoma cell lines. These results will be patented and submitted for publication in the nearest time. A combination of PARP/CHK1 inhibitors with (R)-SKBR-1 remains an open question. Regarding tolerability: auranofin is an established anti-rheumatic drug. PARP inhibitors olaparib and niraparib are also in the clinic. If (R)-SKBR-1 would show favorable properties in preclinical studies, Phase I trials seem reasonable. 

Comment 10: Do the authors expect chronic exposure of NONO-inhibited cells to chronic low-dose chemotherapeutic stress to identify resistance mechanisms (e.g., upregulation of SFPQ, PSPC1)?

Again, this is a subject of our current laboratory research. A very interesting point indeed.

Comment 11: Are there data evaluating the use of patient-derived xenografts (PDX) with varying NONO expression to test efficacy of NONO-targeting drugs and correlate with transcriptional signatures?

We did not identify data on NONO targeting in neuroblastoma xenografts. Floros et al. [70] reported the effects of auranofin on neuroblastoma cell lines and patient-derived xenografts. In this study the role of NONO has not been addressed specifically. Most probably, auranofin attenuates NONO indirectly, therefore the mechanism is questionable. In the revised version we briefly discuss this issue (page 9).

Comment 12: While pharmacological avenues appear to be promising means of targeting NONO, some discussion about how future investigations should strengthen the authors arguments regarding the role of NONO would be beneficial rather than jumping straight to pharmacological strategies. The authors allude to this issue in lines 435-444, by indicating that this may not be sufficient to yield NB cell death. As a result, some suggested experiments which explore the role of NONO would be beneficial. I.e. could the use of CRISPR-Cas9 be beneficial to selectively deplete NONO function in neuroblastoma cell lines (both MYCN-amplified and non-amplified) would allow for researchers to assess viability, transcriptomic shifts, and MYCN target gene expression?

Thank you for raising this important comment. Our intention to switch to pharmacological NONO inhibition is dictated by the clinical need. Stable CRISPR/Cas9-assisted depletion of NONO is supposed to be lethal at least in neuroblastoma cells because its down-regulation by siRNA or (R)-SKBR-1 is cytotoxic (see Kuchur, O.A. et al. Method for enhancing cell death of MYCN-amplified neuroblastoma cells. Russian Federation patent pending, â„– 2025119605, Rospatent, 2025).  More studies are needed to clarify whether the deletion of the NONO gene is critical for survival specifically in neuroblastoma. Another option is a conditional knockout for temporary gene silencing. This is a legitimate experiment to explore.

Comment 13: Replace or expand figure1 with a schematic that summarizes all known NONO interactions in neuroblastoma, including lncRNA binding, protein-protein interactions, its role in DNA repair, and potential therapeutic intervention points. Another option would be a supporting table which summarizes some of the key functionality and supporting references that are used elsewhere in the review.

Figure 1 depicts major NONO interactions. The text contains detailed analysis of these interactions in physiological situations and in cancer. To stay focused on neuroblastoma pathogenesis, we added a new Figure 2 (page 8) summarizing NONO roles in the key neuroblastoma subtypes ADRN and MES. We limited the illustrations to these figures to avoid repetition and not to overload the manuscript.

Comment 14: Improve Table 1 by adding further mechanistic information (e.g., covalent inhibitor, redox modulator, PARP pathway interference).

We added information about mechanisms of action.

Comment 15: Misspelling: first sentence of section 3, the word "pharmacologically" is misspelled as "pharmacalogically." 

Sorry, I do not locate this typo. The term is spelled correctly across the text.

In the revised text we colored major changes, including the new title, in red. Also, please note a couple of crossed-out words and phrases.

Reviewer 2 Report

Comments and Suggestions for Authors

Peer Review Report

Comments for the Authors

Summary

This narrative review explores NONO, an RNA/DNA-binding protein, as a central player in neuroblastoma biology and a promising therapeutic target. The authors do a nice job weaving together NONO's structural features with its functional roles, particularly focusing on neuroblastoma-specific regulatory networks (like the lncUSMYCN→NCYM→MYCN axis and NEAT1/paraspeckle formation). The authors also present several therapeutic approaches, from direct covalent inhibitors to combination strategies. The Simple Summary and Abstract effectively set up why targeting NONO matters for high-risk neuroblastoma patients.

Strenghts

Strong mechanistic foundation: I appreciated how the authors connected NONO's structural domains (RRM1/2, NOPS, coiled-coil regions) to their actual functions. This provides readers with the mechanistic context they need to understand the broader picture.

Neuroblastoma focus stays sharp: The coverage of neuroblastoma-specific mechanisms is comprehensive and current. The discussion of lncUSMYCN/NCYM/MYCN interactions, NEAT1_2-driven paraspeckles with G-quadruplex involvement, NONO's role at super-enhancer-regulated HAND2/GATA2, and the NONO-SOX connections all feel relevant and well-integrated.

Translational potential is clear: The therapeutic section brings together multiple promising approaches - (R)-SKBG-1 as a direct NONO binder, auranofin's effects in MYCN-amplified models, YM155's disruption of NONO-ILF3, and the synthetic lethality concept with PARP+CHK1 combinations. This gives readers concrete paths forward.

Helpful visuals: Figure 1 provides a clean overview of NONO's various roles, and Table 1 nicely summarizes the therapeutic candidates. These will be valuable references for readers.

Limitations / Major Suggestions for Improvement: Help readers understand what matters most

While the comprehensive catalog of NONO interactions is valuable, readers would benefit from understanding which pathways are most important for neuroblastoma. Consider adding a section that ranks these mechanisms by their relevance to specific outcomes (tumor growth, treatment resistance, cell plasticity). For instance, how would the authors prioritize the lncUSMYCN→NCYM→MYCN pathway versus NEAT1 sequestration versus the YY1-PARP1-NONO axis? What experimental evidence supports each ranking? What biomarkers would best capture these different mechanisms in vivo?

Create a practical therapeutic framework: The combination strategies the authors mention are intriguing, but clinicians and translational researchers need more guidance. Consider developing a decision tree or flowchart that shows when to deploy Non-Targeting approaches and how to identify likely responders. Which biomarkers (MYCN/NCYM levels, YY1/PARP1 activity, ferroptosis markers like TfR1/MDA) would guide treatment selection? What resistance mechanisms should we anticipate? What toxicities might emerge? Connecting each drug class to specific pharmacodynamic markers would strengthen the translational impact.

Be transparent about evidence quality: Some key findings, particularly the (R)-SKBG-1 structural work, come from preprints that haven't completed peer review. While these sources can be valuable, readers need to know which claims rest on fully vetted versus preliminary evidence. Authors may consider adding a brief note or table distinguishing peer-reviewed findings from preprints and patents, especially for the compounds listed in Table 1.

Comments on Figures and Tables

Figure 1: This is a strong overview figure. However, authors might enhance it by indicating where ADRN versus MES programs intersect with NONO functions (perhaps ADRN with HAND2/GATA2 and MES with YAP/TAZ pathways).

Table 1: Very practical and useful. To make it even more valuable, consider adding columns for: (1) suggested biomarkers for patient selection, (2) level of neuroblastoma-specific evidence (cell line vs xenograft vs clinical), and (3) expected side effects or drug-drug interactions for combination approaches. Also, clarify whether auranofin's NONO suppression is direct or comes through TRXR inhibition.

References

The citation list is current and appropriate, covering the relevant NONO biology, neuroblastoma mechanisms, and therapeutic approaches.

Areas for improvement:

  • Needs clearer prioritization of mechanisms
  • Would benefit from a biomarker-guided therapeutic roadmap
  • Should distinguish evidence quality levels more explicitly
  • Requires language polishing for clarity

Recommendation

Minor revision recommended

This is a valuable contribution that addresses an important therapeutic target in neuroblastoma. With some reorganization to prioritize mechanisms, a clearer therapeutic framework with biomarkers, more transparent evidence tiering, and language improvements, this will be an excellent resource for the field. The core content is strong - it just needs some refinement to maximize its impact for both basic researchers and clinicians interested in NONO as a therapeutic target.

Comments on the Quality of English Language

Polish the language for clarity: Several passages would benefit from editing to improve readability. The Conclusions section, in particular, has some phrases that obscure rather than clarify the authors' points about druggability challenges and molecular glue hypotheses.

Specific Text Issues: Some examples where the writing could be clearer:

In the Conclusions: "Several difficulties stay on the road to the positive answer" is awkward. Try: "Several challenges remain before we can definitively target NONO therapeutically."

Missing punctuation: "...xenografts in NCG mice This compound elevated..." needs a period after "mice."

Wrong character set: "EGFR и STAT3" uses a Cyrillic character - should be "EGFR and STAT3"

Hyphenation needed: "small molecular weight compounds" should be "small-molecular-weight compounds"

Long sentences in the DDR section would read better if broken up. Consider splitting at natural break points for better flow.

etc.

Author Response

We are grateful to the reviewer for the encouraging criticism.

Comment 1: While the comprehensive catalog of NONO interactions is valuable, readers would benefit from understanding which pathways are most important for neuroblastoma. Consider adding a section that ranks these mechanisms by their relevance to specific outcomes (tumor growth, treatment resistance, cell plasticity). For instance, how would the authors prioritize the lncUSMYCN→NCYM→MYCN pathway versus NEAT1 sequestration versus the YY1-PARP1-NONO axis? What experimental evidence supports each ranking? What biomarkers would best capture these different mechanisms in vivo?

Create a practical therapeutic framework: The combination strategies the authors mention are intriguing, but clinicians and translational researchers need more guidance. Consider developing a decision tree or flowchart that shows when to deploy Non-Targeting approaches and how to identify likely responders. Which biomarkers (MYCN/NCYM levels, YY1/PARP1 activity, ferroptosis markers like TfR1/MDA) would guide treatment selection? What resistance mechanisms should we anticipate? What toxicities might emerge? Connecting each drug class to specific pharmacodynamic markers would strengthen the translational impact.

These questions are really important. However, at the present stage it is premature to unequivocally prioritize NONO-mediated mechanisms of neuroblastoma cell death. Given that NONO is a multifunctional protein, its inactivation is supposed to evoke manifold effects. Current experimental data is insufficient for clear-cut conclusions regarding which pathway is predominant.  More knowledge is necessary to address these pivotal problems and produce a guide for clinicians. The review contains thoughts in regard to drug combinations. However, before a reliable pharmacological NONO inhibitor enters the clinic, and prior to the ultimate proof of target and mechanisms, it seems preliminary to foresee the detailed therapeutic regimens.

We are grateful to the reviewer for raising these themes.  We now discuss them in the section Conclusions, page 13.

Comment 2: Be transparent about evidence quality: Some key findings, particularly the (R)-SKBG-1 structural work, come from preprints that haven't completed peer review. While these sources can be valuable, readers need to know which claims rest on fully vetted versus preliminary evidence. Authors may consider adding a brief note or table distinguishing peer-reviewed findings from preprints and patents, especially for the compounds listed in Table 1.

Agree. Very good point. We decided not to withdraw the research-in-progress sources [71, 75, 76] although in the revised version we stated that these references have not yet been peer-reviewed (legend to Table 1). Nevertheless, we think the readers should be timely informed about the ongoing studies as well. 

Comment 3: Figure 1: This is a strong overview figure. However, authors might enhance it by indicating where ADRN versus MES programs intersect with NONO functions (perhaps ADRN with HAND2/GATA2 and MES with YAP/TAZ pathways).

We agree that more information on NONO in neuroblastoma is worth presenting. We added a new Figure 2 (page 8) showing NONO roles in the biology of the key subtypes ADRN and MES. We decided not to show too many details that require an independent validation. We believe that, at the present state, it is sufficient to claim the directions of thinking. Further research will confirm or criticize the shown mechanisms. 

Comment 4: Table 1: Very practical and useful. To make it even more valuable, consider adding columns for: (1) suggested biomarkers for patient selection, (2) level of neuroblastoma-specific evidence (cell line vs xenograft vs clinical), and (3) expected side effects or drug-drug interactions for combination approaches. Also, clarify whether auranofin's NONO suppression is direct or comes through TRXR inhibition.

We think NONO suppression by auranofin is indirect since TrxR is the established target of this agent (stated in Table 1). To us the effect of auranofin strongly suggests a role of NONO as a redox balance sensor. We now explore this mechanism using combinations of auranofin and specific NONO inactivation [Kuchur, O.A. et al. Method for enhancing cell death of MYCN-amplified neuroblastoma cells. Russian Federation patent pending, â„– 2025119605, Rospatent, 2025; also, a manuscript by our group is in preparation]. At the present stage we limited the knowledge about this effect by mentioning auranofin as a TrxR antagonist in Table 1. We have no direct evidence of auranofin as a NONO binder; structurally, such binding is unlikely.

In accordance with the reviewer’ suggestion we specified individual mechanisms of action for each compound.

Comment 5: The citation list is current and appropriate, covering the relevant NONO biology, neuroblastoma mechanisms, and therapeutic approaches.

When this study was under review, a paper establishing a novel role for NONO in metabolism was published (Zhang S. et al. NONO Maintains SREBP-Regulated Cholesterol Biosynthesis via RNA Binding in Neuroblastoma. FASEB J. 2025, 39(18):e71051. doi: 10.1096/fj.202403267RR). The authors demonstrated that NONO inactivation by siRNA or (R)-SKBG-1 decreased the level of cholesterol in neuroblastoma patient-derived cell lines including Kelly. This important finding further expanded the biological significance of NONO as a key regulatory mechanism in this tumor. It is now mentioned as Notes Added in Proof on page 14 and new ref. [85].

Comment 6: Polish the language for clarity: Several passages would benefit from editing to improve readability. The Conclusions section, in particular, has some phrases that obscure rather than clarify the authors' points about druggability challenges and molecular glue hypotheses.

The section Conclusions and the entire text were thoroughly revised. The issues of druggability were analyzed in this section as well as throughout the text. The molecular glue hypothesis came from ref. [82] in which this developing class is discussed in detail. Interested readers are welcome to peruse this source and refs therein. We decided not to dissect this hypothesis but just mentioned the glues, as well as PROTAC degraders, as an opportunity for medicinal chemists. Could be that these instruments are disadvantageous compared with the covalent inhibitor. This analysis requires new and broad evidence which is beyond the present work. 

Comment 7: In the Conclusions: "Several difficulties stay on the road to the positive answer" is awkward. Try: "Several challenges remain before we can definitively target NONO therapeutically."

Agree. We corrected the sentence as suggested by the reviewer.

Comment 8: Missing punctuation: "...xenografts in NCG mice This compound elevated..." needs a period after "mice."

Done

Comment 9: Wrong character set: "EGFR и STAT3" uses a Cyrillic character - should be "EGFR and STAT3"

Corrected

Comment 10: Hyphenation needed: "small molecular weight compounds" should be "small-molecular-weight compounds"

Corrected

Comment 11: Long sentences in the DDR section would read better if broken up. Consider splitting at natural break points for better flow.

We simplified the phrasing on pages 5-6 to leave the essence unaltered. 

In the revised text we colored major changes, including the new title, in red. Also, please note a couple of crossed-out words or phrases.

Reviewer 3 Report

Comments and Suggestions for Authors

This review article is well written and provides a comprehensive overview of the role of NONO in neuroblastoma development. This review discusses the challenges of drug development to target NONO and highlights recent advances with small molecule inhibitor development for NONO. 

Author Response

Thank you for the encouraging comments!

Reviewer 4 Report

Comments and Suggestions for Authors

This review dives into the role of NONO (non-POU domain-containing octamer-binding protein) as a key player in regulating transcription in neuroblastoma, exploring whether its inactivation could serve as a promising therapeutic approach. The manuscript is both relevant and engaging, especially since pediatric tumors like neuroblastoma are notoriously aggressive and challenging to treat. Targeting transcriptional regulators, particularly those associated with MYCN amplification, represents an exciting possible avenue in oncology. The topic itself is quite fresh. While we know a lot about transcriptional dysregulation in neuroblastoma and the significance of MYCN, spotlighting NONO as a vital transcriptional regulator and potential therapeutic target is less frequently discussed. This review expands the conversation beyond just MYCN regulation, positioning NONO as a central figure in various cellular processes, including transcription circuits, DNA repair, and cell cycle progression. By doing so, it enriches the field by presenting NONO as a multi-dimensional target rather than just a one-dimensional regulator. The text is scientifically dense but clear. It effectively outlines both the biological functions (like normal neurogenesis and transcriptional regulation) and the pathological roles (such as oncogenesis and tumor aggressiveness). The analysis confirms that all citations in the document align with the provided bibliographic references. The citation structure is well-organized and follows a logical sequence, ensuring that the scientific claims in the text correspond appropriately with the supporting sources. Then, the conclusions resonate with the main objective: NONO is pro-oncogenic, and while it poses a challenging target, pharmacological strategies aimed at inhibiting its function show promise. Overall, this text offers a compelling, relevant, and original viewpoint on NONO as both a transcriptional regulator and a therapeutic target in neuroblastoma.

Author Response

We are grateful to the reviewer for favorable evaluation and interest in the study.

Reviewer 5 Report

Comments and Suggestions for Authors

The submitted article gives an overview of the involvement of NONO in the regulation of cellular processes. In it's current form I found that the title is somewhat misleading as most of the article is about different mechanisms of action in various types of cancer. To address  this issue the authors should either consider a title change given the broadness of the submitted material OR consider reducing the general sections and increasing the focus on neuroblastoma related data.

  1. he denoting of genes and their respective proteins needs to be done according to the guidelines and excepted customs (genes in italic, proteins normal). This is missing in the entire paper.
  2. The list of abbreviations used seems to be missing a good deal of abbreviations found in the text.
  3. Several citations are incomplete with only the authors, title and doi listed. Please check and cite according to guidelines

Author Response

Thank you for your comments.

Comment 1: The submitted article gives an overview of the involvement of NONO in the regulation of cellular processes. In it's current form I found that the title is somewhat misleading as most of the article is about different mechanisms of action in various types of cancer. To address this issue the authors should either consider a title change given the broadness of the submitted material OR consider reducing the general sections and increasing the focus on neuroblastoma related data.

We thought about this comment during the writing. Now we suggest a new title (see page 1).

Comment 2:

  1. the denoting of genes and their respective proteins needs to be done according to the guidelines and excepted customs (genes in italic, proteins normal). This is missing in the entire paper.

We checked the notations. Names of genes are italicized whereas proteins are written in straight font.

  1. The list of abbreviations used seems to be missing a good deal of abbreviations found in the text.

We made the list for major abbreviations. Less frequently used terms were mentioned in the text.

  1. Several citations are incomplete with only the authors, title and doi listed. Please check and cite according to guidelines.

We checked References and made corrections. New refs. 83-85 are marked in red.

Round 2

Reviewer 1 Report

Comments and Suggestions for Authors

The authors have appropriately addressed all of my comments/concerns.

Reviewer 2 Report

Comments and Suggestions for Authors

The revised manuscript has been accepted without further changes.

Reviewer 5 Report

Comments and Suggestions for Authors

Authors have completed the requested corrections and the submission can be accepted in present form